# Assessing Construction Constraint Factors on Project Performance in the Construction Industry

Tshweu Given Masoetsa [1], Babatunde Fatai Ogunbayo [1,*], Clinton Ohis Aigbavboa [1] and Bankole Osita Awuzie [2,3]

[1] cidb Centre of Excellence & Sustainable Human Settlement and Construction Research Centre, Faculty of Engineering and the Built Environment, University of Johannesburg, Johannesburg 2006, South Africa
[2] Department of Built Environment, Central University of Technology, Bloemfontein 9301, South Africa
[3] Faculty of Environmental Sciences, KO Mbadiwe University, Ogboko 475102, Nigeria
* Correspondence: babatundeo@uj.ac.za

**Abstract:** A construction constraint is a condition that impedes progress toward meeting construction project goals. This paper seeks to assess the constraint factors affecting project performance in the South African construction industry. The study adopted a quantitative research design, and a questionnaire was designed to retrieve data from the target populations. The target population were construction professionals within the South African construction industry. One hundred and eighty questionnaires were administered to construction professionals within the study area through the purposive quota sampling technique. Retrieved data were analysed using descriptive and exploratory factor analysis. In order to determine the data reliability and the interrelatedness of the variables, Cronbach's alpha test was carried out on each component. The results of the exploratory factor analysis show that stakeholders' inappropriate project scheduling and coordination factors, organisation and government policies factors, and organisation and government policies factors were the leading constraints affecting construction project performance in the South African construction industry. Due to time and distance constraints, this study was limited to construction professionals in South Africa's Free State province. The paper concluded that to reduce the construction constraints affecting construction project performance, construction professionals must improve their project scheduling, coordination, organisational policies, and managerial capacity. The paper's findings will assist stakeholders in identifying and overcoming construction constraints in construction projects' execution and delivery.

**Keywords:** constraints; construction projects; construction industry; free state; South Africa

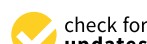



## 1. Introduction

The construction industry is one of the major sectors responsible for the economic growth of nations due to its importance and contributions to Gross Domestic Product [1,2]. The diversity of the construction industry includes the nature of its operations, involvement of different stakeholders, different construction phases, applicable codes, bylaws, and regulations in ensuring successful project planning, designing, construction, and delivery [3–5]. Similarly, refs. [6–9] point out that the success of a construction project is determined by appropriate timing and budgeting for construction projects in line with the quality specified. Refs. [10,11] argue that stakeholders' involvement and how the project is scheduled in terms of cost, quality, and time determine its success or failure. Refs. [12,13] opine that the construction industry encounters numerous constraints during the project life cycle ranging from overruns of time, cost, project scope, conflicts, and low-quality performance.

Construction constraints, as defined by refs. [14,15], are conditions or forces that obstruct the progress of construction operations toward achieving the triple project's objectives or goals of cost, time, and quality, which is considered the criterion for project

success. Ref. [16] defines a constraint in project management as a point where the project fails to perform as expected. Ref. [17] asserts that the construction industry's paradigm shift due to the sites' complex nature, duration performance, imbalanced cash flows, and complex contractual agreements pose several constraints that usually prevent construction projects from being completed on time and within the cost budgets. Refs. [14,18] posit that stakeholders must have studied the possibility of project constraints at the outset by understanding the dynamics of the project and how different constraints are interrelated. Thus, identifying these constraints will provide the practical steps for making effective organisational decisions that will reduce the impact of these constraints in the construction process [19].

However, refs. [20,21] posit that construction project execution is risky and prone to certain constraints and circumstances threatening successful project delivery. Ref. [16] argues that controlling such constraints is a precondition for the astonishing performance of the construction project. Ref. [22] sustains that the consequence of construction constraints are drawbacks to project success leading to disputes or litigations among the stakeholders, which can affect the project performance. In their study, ref. [23] noticed that constraints could involve all the parties to a contract. If not well-managed, it will affect the contractual relations among the stakeholders. Therefore, the negative impacts of constraints on construction projects' operation include time and cost overruns, a decline in profit and productivity, and damage to contractual relations [24]. Refs. [25,26] state that constraints must be identified and managed at the early stage of the project because ignoring them could lead to conflicts, disputes, and sabotage of stakeholders' relationships. As such, constraints not identified and managed may produce controversies among stakeholders and directly or indirectly cost the clients and contractors, compromising project quality and scope during the design and implementation ref. [27].

Conversely, ref. [28] posits that insufficient previous research focuses on the construction industry's emergence of conflicts and project constraints. The factors causing construction constraints and their relative impacts on construction operations are rarely known or fully explored. Thus, the rapid rate of development and construction operations in the construction industry demands a research study focusing on identifying factors causing constraints and how they affect project goals' performance and delivery. Hence, this study aims to bridge the knowledge gap by assessing factors causing constraints in construction project performance in the construction industry, using construction projects in South Africa as a case study.

Although a previous study of ref. [29] aimed to assess the criticality of the factors that influence the performance of highway projects, the study cannot be generalised for general construction work because it focuses more on highway projects. Highway projects cannot be entirely compared with other construction projects, given the technicality involved in highway projects. Hence, this study is carried out to focus more on general construction projects. Further, the previous study of ref. [29] was carried out in another developing country, whereas what is applicable in their construction project process might not be applicable in the South African context. These submissions necessitate carrying out this study.

This study precisely focused on assessing factors causing constraints to construction project performance in the construction industry. The motivation for conducting this study in the South African construction industry is that previous studies (refs. [16,23]) on construction constraints only focused on consultants and contractors within construction projects. This study will fill this gap by assessing factors causing constraints in construction project performance among construction professionals within their different professional bodies in the South African construction industry.

The study is important for construction industry professionals in identifying construction constraints early toward achieving a more effective construction project performance. The review of prior literature, theoretical background, methodology, results, discussion of findings, conclusion, and recommendations are among the sections of the article that aid in reaching the stated purpose.

## 2. Literature Review

### 2.1. Findings from Extant Literature on Factors Causing Construction Constraints

Construction projects succeed when the deliverable objectives are attained within the time, budget, quality, and safety progress of construction operations [30]. The study of constraints on resources, risk, and management practices in construction projects cannot be ignored because of its influence on project deliverables [28]. The paradigm shift in construction project management emphasises the importance of identifying major constraint factors against the traditional concept of the triple constraints of time, cost, and quality [31]. The construction project objectives (time, cost, and quality) are limiting factors that can impact project delivery, quality, and overall project success if not well-managed [30]. Hence, in Pakistan, ref. [32] identified "natural disaster", "financial and payment difficulties", "poor planning", "poor site management", "insufficient stakeholders experience", and "shortage of materials and equipment" as the major factors causing construction constraints in project execution. Additionally, ref. [28] investigated the effect of stakeholders' conflicts on projects constraints and attributes of the factors causing constraints to "lack of communication", "poor quality delivery", "change in design and rework", "safety regulations", "workers productivity", and "protection of the environment".

Ref. [33] posited the six factors particular to Thailand's construction industry as constraints relating to owners' problems, designers, construction management related issues, contractors, and resources suppliers. The findings of ref. [34] on the Iranian construction operatives' productivity present factors causing project constraints as "lack of material", "design deficiency and/or change order", "lack of proper tool and equipment", "equipment breakdown", and "weather and site condition". Ref. [35] conducted a study on stakeholders' assessment of constraints to project delivery. The study aimed to identify and assess the constraints to construction project delivery and identified fifty factors causing constraints, mostly in developing countries. Thus, the study attributed the most frequent constraints on project delivery to "poor communication", "lack of coordination and conflicts between stakeholders", "weather/climate conditions", "ineffective or improper planning", "material shortages", "financial problems", "payment delays", "equipment/plant shortage", "lack of qualified stakeholders", "labour shortages", and "poor site management".

Similarly, the study of ref. [16] showed that constraint factors that have high impacts on construction project performance include "improper allocation of funds to parties", "land acquisition", "building regulation", "safety regulation", "dispute in the contractual agreement", "government labour laws", "delay in solving design issues", "inappropriate project cost estimate", and "flawed drawing and details". In Nigeria, ref. [36] identified four external environmental factors causing constraints and affecting project performance: land acquisition tussles, weather conditions, economic situation, and government policies. Further, ref. [37] grouped the factors under political, legal, construction techniques and resources, economic and financial, sociocultural, and physical. Ref. [38] identified a shortage of cash flow, clients' financial difficulties, and poor procurement. Ref. [39] argued that project constraints that influence productivity include a lack of incentives system, poor health conditions of workers, material delay, inadequate site amenities, and an aging workforce.

Ref. [40] attributed factors causing constraints in the Congolese construction industry to client and management, production scheduling and contract, shortages of experience stakeholders and skilful design team, and client/owner payment delay. Similarly, ref. [41] identified delays in assessing changes in the scope of work by the consultant, contractor financial misappropriation, shortage of contractor's experience, design errors, and inadequate site investigation. In South Africa, ref. [42] posited disputes related to contractual documents, undocumented changes in design, financial difficulties, unbalanced cost of materials, high cost of equipment, poor communication between design and construction team, economic instability, and forex exchange rate as the factors causing construction constraints. Ref. [43] posited that constraints in the South African construction industry

always lead to variations, overrun costs, and times due to changing project scope, financial difficulties, government policies and regulations, and inappropriate project cost estimation.

Refs. [42,43] aimed to investigate the causes of cost overruns and management of cost constraints, which is just one of the key elements of the traditional concept of triple constraints in construction projects. Findings from both studies were based on contractors' and consultants' perspectives using descriptive statistical procedures. However, the contributions of this study differ from both studies because they assess the factors causing constraints in construction project performance from the angle of the traditional concept of triple constraints in construction projects based on construction professionals' perspectives. Further, retrieved data in this study are analysed using descriptive and exploratory factor analysis.

*2.2. Theoretical Background*

In explaining the factors causing constraints affecting the project performance, the theory of constraints (TOC) model [44] was engaged. The theory of constraints (TOC) explains five steps to identify and eliminate organisation constraints. The theory of constraints provides the organisation with a logical philosophy of continuous improvement, identifying factors limiting the organisation from attaining goals. Ref. [44]'s theory of constraints' first steps involves identification in the organisation system, harnessing the system constraints, subordinate strategic and tactical approach to the above decision, elevating the system constraints, and reapplying the procedures for continuous improvement. Ref. [45] detailed the evolution of TOC and its application in five different eras citing ref. [44], namely, the first era focused on optimisation production technology 1979–1984; followed by the second era, named the goal era, 1984–1990; the third era was named haystack syndrome 1990–1994; the fourth era was tagged it's not locked era 1994–1997; and the fifth era is called critical chain era 1997–2000. According to refs. [46–48], TOC has grown interested in different fields over the years. Its application has cut across the fields of accounting, marketing, logistic, construction project management, and many industries that desire system change.

Refs. [49,50] posited that the theory of constraints applies to construction project management. Ref. [49] explained that the TOC deals with the fundamental aspects of construction project management, including exercising control, monitoring the project's input and output, and evaluating and selecting the best alternatives to ensure client satisfaction. Ref. [48] posited that constraints limit an organisation's performance; identifying and eliminating constraints is the focus of TOC because of its continuous search for improving the organisation system by dealing with the constraints. Similarly, ref. [51] suggested that the theory of constraints applies to construction project management because its operations are similar to the manufacturing production process in which productivity is affected by various constraints.

Thus, ref. [51] combined the factors causing construction project management constraints under five categories: environmental, economic, legal, technical, and social. With this understanding, this study is underpinned by the theory of constraints and its application in assessing factors causing construction constraints in the project performance using construction projects in Free states province, South Africa, as a case study.

Subsequently, as shown in Table 1 above, a synthesis in the current study of the views developed by various authors provides a more holistic outline to guide this study. Thus, detailed in Table 1 are the factors causing constraints on construction projects that guide this study as extracted from relevant literature. As seen in the literature reviewed, the factors represented the views different studies have advocated primarily on constraints in construction projects.

**Table 1.** Factors causing constraints on construction projects.

| Codes | Factors Causing Constraints (FCA) | Authors |
|---|---|---|
| FCA 1 | Climate change resilience | Refs. [21,23,28,31,34] |
| FCA 2 | Inappropriate project cost estimation | Refs. [16,20,26,33] |
| FCA3 | Traditional beliefs of people | Refs. [27,30,33,36] |
| FCA4 | Lack of supervision onsite | Refs. [16,20,28,39] |
| FCA5 | Delay in materials supply | Refs. [28,31,32,34,40] |
| FCA6 | Poor coordination stakeholders | Refs. [16,28,34,37] |
| FCA7 | Construction workers strikes | Refs. [21,27,33,38] |
| FCA8 | Poor communication | Refs. [28,31,34,36,41] |
| FCA9 | Ownership financial problems | Refs. [3,27,32,37] |
| FCA10 | Poor provision of equipment | Refs. [20,31,32,34] |
| FCA11 | Not completing the project as planned | Refs. [26,28,37] |
| FCA12 | Building regulations | Refs. [16,29,37,40] |
| FCA13 | Safety regulations | Refs. [21,28,33,38] |
| FCA14 | Changes in drawings/design | Refs. [20,23,28,30] |
| FCA15 | Design for deconstruction and disposal | Refs. [20,26,28,39] |
| FCA16 | Waste, water management, dust, vibration, and noise | Refs. [23,26,31,38] |
| FCA17 | Poor planning and scheduling | Refs. [24,31,33,34] |
| FCA18 | Difficulties in obtaining work permits | Refs. [22,28,31,42] |
| FCA19 | Land acquisition | Refs. [27,35,38,39] |
| FCA20 | Availability of local workforce | Refs. [8,16,20,26] |
| FCA21 | Work laws (of the current government) | Refs. [15,25,35,39] |
| FCA22 | Dispute related to contractual documents | Refs. [12,18,26,28,33] |
| FCA23 | Use of inexperienced workers | Refs. [8,23,31,34,38] |
| FCA24 | Delay in solving design problems | Refs. [20,28,31,34] |
| FCA 25 | Difficulties in obtaining loans from financiers | Refs. [8,26,31,38] |
| FCA 26 | Air, water, or ground pollution | Refs. [16,28,31,34] |
| FCA 27 | Usage of sustainable materials | Refs. [21,28,31,32,34] |
| FCA 28 | Preservation of ecology and transportation | Refs. [23,32,37,40] |
| FCA 29 | Improper allocation of money to contractors | Refs. [20,31,34,38,39] |

Source: Author's compilation (2022) as reviewed from the literature.

## 3. Methodology

As indicated in Figure 1, a quantitative research method was adopted to investigate factors causing constraints in construction project performance in the South African construction industry. The respondents targeted construction professionals in the built environment: architects, construction managers, consultants, engineers, project managers, site agents, quality coordinators, and quantity surveyors. The choice of using these construction professionals was based on their involvement in building construction projects from planning, designing, construction delivery, and administration in the study area. Hence, a closed-ended questionnaire was designed in line with variables from the extant literature reviewed in Table 1.

The survey requested respondents to indicate their level of agreement with each identified factor causing constraints in the construction projects in Free state, South Africa. Free-State province was chosen for the study because of different construction projects (government and private) sited within the province with a high rate of abandonment [52]. A five-point Likert scale was used: 1 = Strongly disagree (SD), 2 = Disagree (D), 3 = Neutral (N), 4 = Agree (A), 5 = Strongly agree (SA). Through the purposive quota sampling technique, one hundred and eighty (180) questionnaires were administered to construction professionals within the study area. Retrieved data were analysed using descriptive and exploratory factor analysis (EFA). Out of the administered questionnaires, a total of one hundred and fifty (150) copies were retrieved from the respondents for the analysis. This represents 83.33% of the total questionnaires administered. As postulated by ref. [53], a total of 27,000 construction professionals worked within the Free-State province construction industry of South Africa. Ref. [54]'s equation, as cited by [55], was used to calculate a sample size that can represent the total population of 27,000 construction professionals in

the Free-State province construction industry of South Africa. Equation (2) below is used to calculate the sample size for this research.

$$n = N/[1 + N(e)^2] \qquad (1)$$

where $n$ = the random sample size, $N$ = the population size, and $e$ = the level of precision.

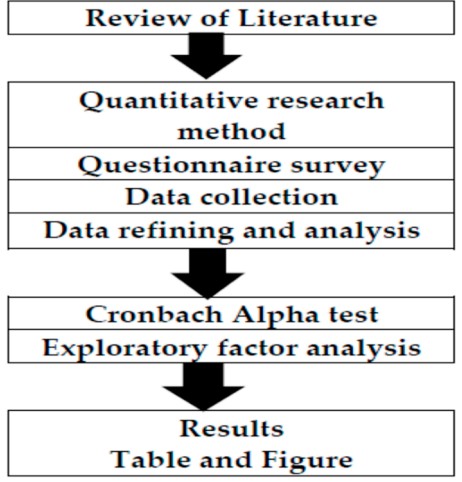

**Figure 1.** Research methodology adopted for the study.

According to ref. [54], the level of precision is the 95% confidence level and is equivalent to $p = 0.05$, and it was adopted for this equation in this research. Therefore, the sample size n for this research is

$$n = 27{,}000/[1 + (27{,}000) \times (0.05)^2] = 68.5 \qquad (2)$$

From Equation (1), the sample size required for this study is 68.5 respondents. However, the number of respondents used for the study is 150, representing 18.5% of the total population of 27,000. This is an acceptable value required for using exploratory factor analysis (EFA), as [56] suggested, which recommended a sample size of at least 100 respondents for effect analysis. Ref. [56] stated further that it is trusted that the bigger the sample size, the lower the probability of blunder in summing up the populace. This study's sample size of 150 respondents is also adequate based on the recommendations of [57]. Statistical package for the social sciences (SPSS) version 21 was used to analyse the data obtained from the field survey. This was followed by descriptive analysis using percentage, frequency, standard deviation, and ranking. The data adequacy for exploratory factor analysis (EFA) was determined through Kaiser–Meyer–Olkin (KMO) and Bartlett's sphericity test. Ref. [58] posited that EFA helps researchers reduce large data to smaller numbers by exploring their level of relationship. Cronbach's alpha test was conducted to determine the data reliability and the interrelatedness of the variables in each component. Ref. [59] opined that Cronbach's alpha test explores the scale reliability of data via their internal consistency. In addition, ref. [60] stated that for data reliability, the coefficient of Cronbach's alpha scale must return a 0.7 value minimum, justifying the reliability of the data collection instrument; the results of the analysis were presented in figures and tables. The EFA method used in this study to address the factors causing constraints in construction project performance is distinctive. The benefit of using EFA for this study was that it aided in identifying groups of interrelated variables (constraint factors) to see how they are related to each other.

## 4. Results and Discussion of Findings

### 4.1. Demographic Information of the Respondents

Figure 2 shows the years of working experience of the respondents. Six percent of the total respondents had less than two years of work experience, followed by 20% with less than five years of work experience. However, 22% of the respondents had years of working experience ranging from six to ten years, and 22% had years of working experience ranging from eleven to fifteen years, respectively. Nineteen percent had an experience that ranged from sixteen to twenty years, and 8% had an experience that ranged from twenty-one to twenty-five years. Lastly, 3% of the respondents had more than 25 years of industry working experience. The data analyses of the respondents' years of experience justified that the respondents were experienced enough to respond to survey questions that will be asked in the subsequent sections of the questionnaire data analyses.

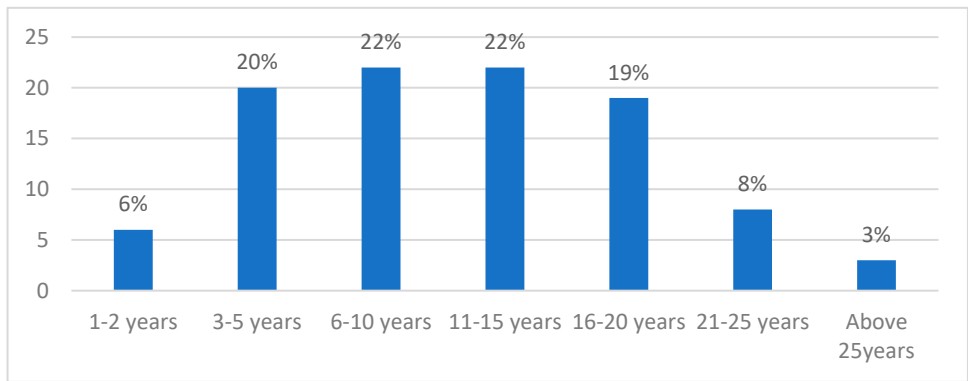

**Figure 2.** Respondents' years of experience.

Figure 3 shows findings relating to respondents' professions, which revealed that 7% were project management, 3% consultants, followed by 37% contract managers, 23% engineers, 5% architects, 18% quantity surveyors, 4% site agents, and 3% quality coordinators.

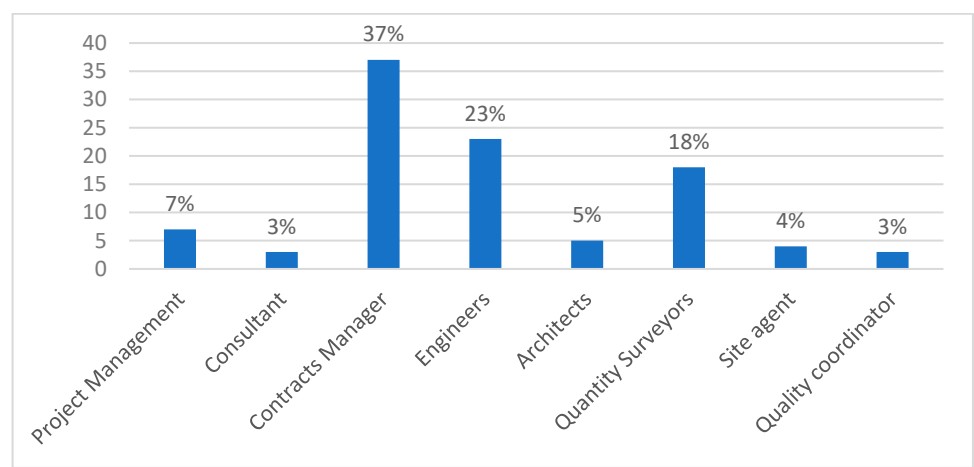

**Figure 3.** Respondents' professional qualification.

Figure 4 presents the educational qualification of the respondents. Three percent of the respondents had a national diploma, 5% had a doctoral degree, 17% had a professional degree, 19% had an honour degree, 24% had a bachelor's degree, and 32% had a master's degree. These ratios justify that the respondents involved in this study had the required levels of education for this study.

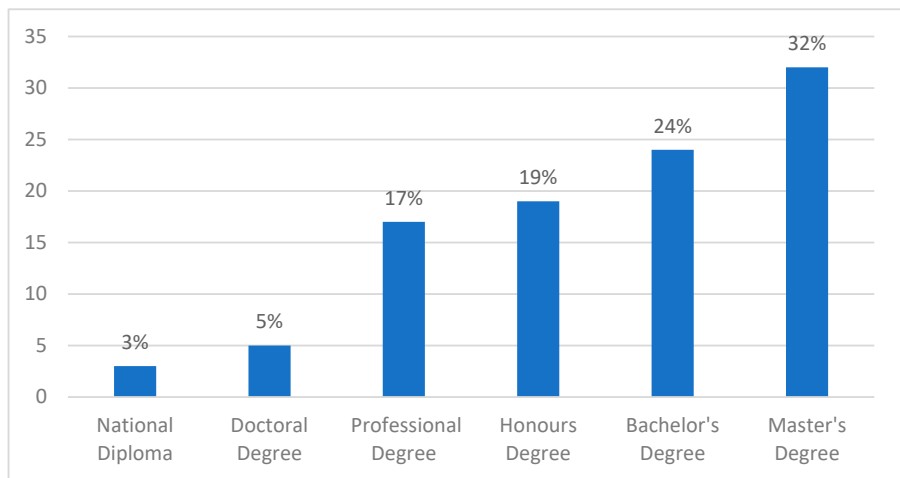

**Figure 4.** Respondents' educational qualification.

*4.2. Results from the Exploratory Factor Analysis*

4.2.1. KMO and Bartlett's Test

Table 2 presents the result of EFA measures of data sample adequacy using Kaiser–Meyer–Olkin (KMO) and Bartlett's test of sphericity. The KMO returned a value of 0.826 more than the 0.6 set as minimum criteria, and Bartlett's test returned a significant value of 0.000 below 0.5 as benchmarked data suitability for factor analysis [59].

**Table 2.** KMO and Bartlett's Test.

| KMO and Bartlett's Test | | |
| --- | --- | --- |
| Kaiser–Meyer–Olkin Measure of Sampling Adequacy | | 0.826 |
| Bartlett's Test of Sphericity | Approx. Chi-Square | 2713.031 |
| | Df | 406 |
| | Sig. | 0.000 |

4.2.2. Scree Plot

Similarly, Figure 5 shows the scree plot for the data set, highlighting the eigenvalues for all the 29 variables of factors causing constraints (FCA) analysed. The scree plot shows that only six factors are above 1 on the eigenvalue axis [57,58]. Further inspection of the scree plot reveals that the last significant break on the plot was on the sixth factor, which confirms the extraction of six factors. The steeper portion of the slope shows the large factors, while the gradual trailing off shows the rest of the factors that have an eigenvalue lower than 1.

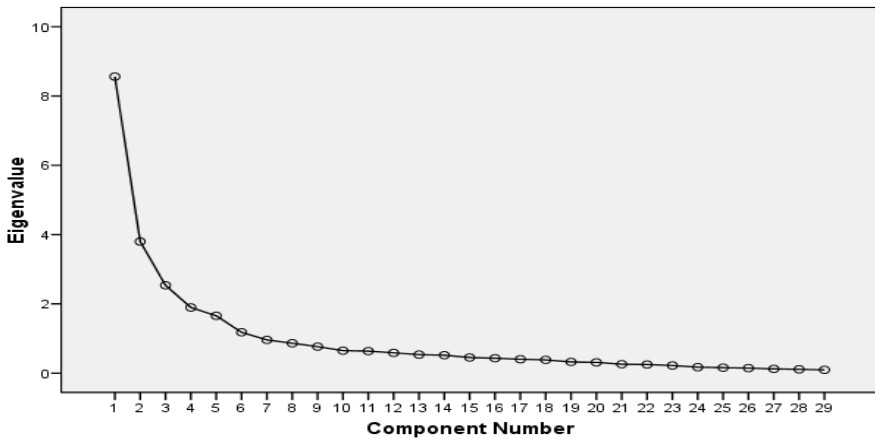

**Figure 5.** Scree plot.

### 4.2.3. Communalities

Table 3 shows the various FCA items after extraction, which should contain values above 0.1. The values as seen in the table show that all the FCA variables have extraction values greater than 0.1 and are, therefore, suitable for factor analysis.

**Table 3.** Communalities for FCA variables.

| FCA Variables | Initial | Extraction |
| --- | --- | --- |
| Delay in solving design problems | 1.000 | 0.630 |
| Inappropriate project cost estimation | 1.000 | 0.658 |
| Poor communication | 1.000 | 0.537 |
| Poor planning and scheduling | 1.000 | 0.514 |
| Poor coordination between different agencies | 1.000 | 0.633 |
| Not completing the project in a given duration | 1.000 | 0.732 |
| Difficulties in obtaining loans from financiers | 1.000 | 0.709 |
| Improper allocation of money to related parties | 1.000 | 0.713 |
| Availability of local workforce | 1.000 | 0.627 |
| Land acquisition | 1.000 | 0.823 |
| Safety regulations | 1.000 | 0.790 |
| Building regulations | 1.000 | 0.836 |
| Difficulties in obtaining work permits | 1.000 | 0.796 |
| Dispute related to contractual documents | 1.000 | 0.611 |
| Changes in drawings/design | 1.000 | 0.597 |
| Work laws (of the current government) | 1.000 | 0.621 |
| Air, water, or ground pollution | 1.000 | 0.744 |
| Usage of hazardous or sustainable materials | 1.000 | 0.810 |
| Use of inexperienced workers | 1.000 | 0.646 |
| Waste and water management, dust, vibration, and noise | 1.000 | 0.649 |
| Preservation of ecology, traffic, and transportation | 1.000 | 0.629 |
| Climate change resilience | 1.000 | 0.626 |
| Design for deconstruction and disposal | 1.000 | 0.581 |
| Traditional beliefs of people | 1.000 | 0.598 |
| Ownership financial problems | 1.000 | 0.698 |
| Construction workers strikes | 1.000 | 0.668 |
| Delay in materials supply | 1.000 | 0.766 |
| Poor provision of equipment | 1.000 | 0.721 |
| Lack of supervision on site | 1.000 | 0.666 |

Extraction Method: Principal Component Analysis.

### 4.2.4. Total Variance Explained

Table 4 shows the eigenvalues of the variables in the data set. The Kaiser's criterion, which entails retaining factors with eigenvalues that are above 1.0, was employed. Hence, six factors with eigenvalues greater than 1 were retained. The eigenvalues of the retained factors are 8.561, 3.800, 2.536, 1.896, 1.656, and 1.180, which explains 11.063%, 9.078%, 8.001%, 6.751%, 6.470%, 6.127%, 5.633%, and 5.396% of the variances, respectively. These six clusters of factors represent 67.691% of the total variance, which highlights the importance of all 29 variables measured.

### 4.2.5. Pattern Matrix[a]

Table 5 presents the pattern matrix, highlighting how the factors have been clustered together. The table shows that the exploratory factor analysis returned six components of factors causing constraints in construction project performance. The table also revealed the arrangement of the variable under each returned component according to their significance. After a critical study of Table 5, a common name for each clustered component was decided [55,56]. Factor 1 is named "stakeholders' inappropriate project scheduling and coordination"; factor 2, "organisation and government policies"; factor 3, "ownership financial and contractual irregularity"; factor 4, "external factors"; factor 5, "Project peculiarity factors"; factor 6, "managerial factor".

**Table 4.** Total variance explained.

| Component | Initial Eigenvalues | | | Extraction Sums of Squared Loadings | | | Rotation Sums of Squared Loadings [a] |
|---|---|---|---|---|---|---|---|
| | Total | % of Variance | Cumulative % | Total | % of Variance | Cumulative % | Total |
| FCA 1 | 8.561 | 29.521 | 29.521 | 8.561 | 29.521 | 29.521 | 6.927 |
| FCA 2 | 3.800 | 13.105 | 42.625 | 3.800 | 13.105 | 42.625 | 4.848 |
| FCA 3 | 2.536 | 8.746 | 51.371 | 2.536 | 8.746 | 51.371 | 3.325 |
| FCA 4 | 1.896 | 6.538 | 57.909 | 1.896 | 6.538 | 57.909 | 4.632 |
| FCA 5 | 1.656 | 5.712 | 63.621 | 1.656 | 5.712 | 63.621 | 3.457 |
| FCA 6 | 1.180 | 4.070 | 67.691 | 1.180 | 4.070 | 67.691 | 1.625 |
| FCA 7 | 0.960 | 3.312 | 71.003 | | | | |
| FCA 8 | 0.864 | 2.979 | 73.982 | | | | |
| FCA 9 | 0.766 | 2.641 | 76.623 | | | | |
| FCA 10 | 0.650 | 2.242 | 78.866 | | | | |
| FCA 11 | 0.637 | 2.197 | 81.063 | | | | |
| FCA 12 | 0.587 | 2.025 | 83.088 | | | | |
| FCA 13 | 0.537 | 1.851 | 84.939 | | | | |
| FCA 14 | 0.519 | 1.790 | 86.730 | | | | |
| FCA 15 | 0.452 | 1.559 | 88.289 | | | | |
| FCA 16 | 0.431 | 1.487 | 89.776 | | | | |
| FCA 17 | 0.402 | 1.388 | 91.164 | | | | |
| FCA 18 | 0.384 | 1.323 | 92.487 | | | | |
| FCA 19 | 0.325 | 1.121 | 93.608 | | | | |
| FCA 20 | 0.312 | 1.077 | 94.685 | | | | |
| FCA 21 | 0.259 | 0.892 | 95.577 | | | | |
| FCA 22 | 0.250 | 0.863 | 96.440 | | | | |
| FCA 23 | 0.221 | 0.763 | 97.203 | | | | |
| FCA 24 | 0.174 | 0.602 | 97.804 | | | | |
| FCA 25 | 0.159 | 0.549 | 98.353 | | | | |
| FCA 26 | 0.146 | 0.504 | 98.857 | | | | |
| FCA 27 | 0.124 | 0.428 | 99.285 | | | | |
| FCA 28 | 0.111 | 0.382 | 99.667 | | | | |
| FCA 29 | 0.096 | 0.333 | 100.000 | | | | |

Extraction Method: Principal Axis Factoring. [a] When factors are correlated, sums of squared loadings cannot be added to obtain a total variance.

**Table 5.** Pattern matrix[a].

| Variables | Pattern Matrix[a] Component | | | | | |
|---|---|---|---|---|---|---|
| | 1 | 2 | 3 | 4 | 5 | 6 |
| Not completing the project in each duration | 0.903 | | | | | |
| Difficulties in obtaining loans from financiers | 0.812 | | | | | |
| Availability of local workforce | 0.802 | | | | | |
| Improper allocation of money to related parties | 0.771 | | | | | |
| Poor communication | 0.732 | | | | | |
| Inappropriate project cost estimation | 0.704 | | | | | |
| Delay in solving design problems | 0.682 | | | | | |
| Poor coordination between different agencies | 0.593 | | | | | |
| Poor planning and scheduling | 0.453 | | | | | |
| Safety regulations | | 0.912 | | | | |
| Building regulations | | 0.913 | | | | |
| Land acquisition | | 0.892 | | | | |
| Difficulties in obtaining work permits | | 0.873 | | | | |
| Dispute related to contractual documents | | 0.774 | | | | |

**Table 5.** *Cont.*

| Variables | Pattern Matrix[a] Component | | | | | |
|---|---|---|---|---|---|---|
| | 1 | 2 | 3 | 4 | 5 | 6 |
| Traditional beliefs of people | | | 0.754 | | | |
| Changes in drawings/design | | | 0.672 | | | |
| Ownership financial problems | | | 0.663 | | | |
| Construction workers strikes | | | 0.654 | | | |
| Poor provision of equipment | | | 0.543 | | | |
| Delay in materials supply | | | 0.534 | | | |
| Climate change resilience | | | | 0.711 | | |
| Use of inexperienced workers | | | | 0.684 | | |
| Design for deconstruction and disposal | | | | 0.662 | | |
| Preservation of ecology, traffic, and transportation | | | | 0.606 | | |
| Work laws (of the current government) | | | | 0.554 | | |
| Usage of hazardous or sustainable materials | | | | | 0.873 | |
| Air, water, or ground pollution permit | | | | | 0.782 | |
| Waste and water management, dust, vibration, and noise permit | | | | | 0.732 | |
| Lack of supervision on site | | | | | | 0.823 |

Extraction Method: Principal Axis Factoring. Rotation Method: Oblimin with Kaiser Normalisation. [a] Rotation converged in 15 iterations.

**Component one: stakeholders' inappropriate project scheduling and coordination**.

As shown in Table 5, the first component had nine variables loaded into the component: "not completing the project in each duration (90%)", "difficulties in obtaining loans from financiers (81%)", "availability of local workforce (80%)", "improper allocation of money to related parties (77%)", "poor communication (73%)", "inappropriate project cost estimation (70%)", "delay in solving design problems (68%)", "poor coordination between different agencies (59%)", and "poor planning and scheduling (45%)". Thus, this cluster gathered 29.521% of the total variance. The factors loaded in the first component emphasised the stakeholder's inappropriate project scheduling and coordination, which directly impact the overall performance of construction projects. The findings are in line with refs. [20,28,32,34,35], which opine that inappropriate project cost estimation, improper allocation of money to related parties, and delay in solving design changes cause constraints to stakeholders in construction project management. This makes the factors loaded in the first component important in determining successful project performance and delivery.

**Component two: Organisation and government policies**.

As shown in Table 5, the second cluster had five variables loaded into the component: "Safety regulations (91%)", "building regulations (91%)", "land acquisition (89%)", "difficulties in obtaining work permits (87%)", and "dispute related to contractual documents (77%)". Thus, this cluster gathered 13.105% of the total variance. These factors address the external and internal policies that govern the construction industry operations. The findings confirm the studies of refs. [16,30,36,39] that lack of adherence to government regulations regarding safety, land acquisition, and contractual dispute-related issues are factors causing construction constraints that affect project performance in the construction industry.

**Component three: Ownership financial and contractual delays**.

As shown in Table 5, the third cluster had six variables loaded into the component: "traditional beliefs of people (75%)", "changes in drawings/design (67%)", "ownership financial problems (66%)", "construction workers strike (65%)", "poor provision of equipment (54%)", and "delay in materials supply (53%)". The factors loaded in the third component show that clients' decisions contribute to factors that cause construction constraints that affect projects' performance in the construction industry. The cluster gathered 8.746% of the total variance. The finding agrees with [12,38,43] studies that financial problems of the client, poor equipment provisions, delays in material supply, worker strikes,

and traditional beliefs are key factors causing construction constraints affecting project performance in the construction industry.

**Component four: External factors.**

As shown in Table 5, the fourth cluster had five variables loaded into the component: "climate change resilience (71%)", "use of inexperienced workers (68%)", "design for deconstruction and disposal (66%)", "preservation of ecology, traffic, and transportation (61%)", and "work laws of the current government (55%)". Hence, this cluster gathered 6.538% of the total variance. The findings align with those of refs. [8,23,32,35,39] that external factors such as climate change resilience, preservation of ecology, and poor transportation system are factors causing construction constraints leading to poor project performance within the construction industry.

**Component five: Project peculiarity factors.**

As shown in Table 5, the fifth cluster had three variables loaded into the component: "permit on the usage of hazardous or sustainable materials (87%)", "air, water, or ground pollution permit (78%)", and "waste and water management, dust, vibration, and noise permit (73%)". The three factors refer to the requirement of local enforcement agencies to control the use of natural resources, hazardous materials, and the management of construction pollution. Thus, the cluster gathered 5.712% of the total variance. The findings agree with refs. [22,28,32,35] that factors such as the use of hazardous materials, poor waste management, and delay in the issuance of permits cause construction constraints that affect project performance in the construction industry.

**Component six: Managerial factors.**

As shown in Table 5, the sixth component had one variable loaded into the component: "lack of supervision onsite (82%)". The cluster gathered 4.070% of the total variance. The finding is in line with refs. [28,32,35] that poor supervision is a factor causing construction constraints affecting construction performance in the construction industry.

4.2.6. Component Correlation Matrix and Reliability of the Factors

From Table 6, the relationship between the cluster groups is shown in the component correlation matrix in that some of the clusters have values around 0.300. This is an indication that there is a relationship between these clusters. Moreover, the variables of the components within the factors highly correlate with each other. It also shows a relationship and dependence among the variables [59,60]. Additionally, the Cronbach alpha coefficient test conducted on each variable in Table 5 shows a value of between 0.711–0.917. The result indicates that the variables measured are reliable and valid and that the data collection instrument used is reliable in collecting information [59,60].

**Table 6.** Component Correlation Matrix and Reliability of the factors.

| Factor | 1 | 2 | 3 | 4 | 5 | 6 | Cronbach's Alpha Coefficient |
|---|---|---|---|---|---|---|---|
| Component 1 | 1 | −0.29 | 0.182 | 0.374 | 0.2 | 0.141 | 0.913 |
| Component 2 | −0.29 | 1 | −0.06 | −0.24 | 0.03 | −0.11 | 0.917 |
| Component 3 | 0.182 | −0.06 | 1 | 0.122 | 0.22 | 0.08 | 0.789 |
| Component 4 | 0.374 | −0.24 | 0.122 | 1 | 0.06 | −0.03 | 0.802 |
| Component 5 | 0.202 | 0.028 | 0.22 | 0.06 | 1 | 0.092 | 0.711 |
| Component 6 | 0.141 | −0.11 | 0.08 | −0.03 | 0.09 | 1 | 0.854 |

Extraction Method: Principal Axis Factoring. Rotation Method: Oblimin with Kaiser Normalisation.

**5. Conclusions and Recommendations**

This study assessed factors causing construction constraints in project performance in the construction industry. The study adopts EFA to explore the significance of the twenty-nine factors identified from the review of the literature. The EFA returned six components of constraint factors: "stakeholders' inappropriate project scheduling and

coordination factors", "organisation and government policies factors", "ownership financial and contractual irregularity factors", "external factors", "project peculiarity factors", and "managerial factors".

The factors included in the components explain the construction constraints causing constraints on project performance and delivery in the construction industry. The study's findings identify the underlying relationship between measured constraints' variables from literature. Thus, the theoretical assessment is consistent with the research's empirical outcomes. Further, the feedback corresponds with the literature on factors causing constraints in construction project performance based on the traditional concept of triple constraints in construction projects. The study concludes that the identified factors causing constraints on construction projects in the six components directly influence project goals, performance, and delivery.

The study findings add to the knowledge gap by identifying particular factors causing constraints on project performance in the construction industry. Moreover, due to the difference in location and scope of work, the grouping of the factors causing constraints ends up in six components, compared with a grouping of four components by an earlier study. The implications of the study's findings show that constraints in construction projects as identified in the six components will result in costs and time overruns, delays, disputes, and litigations, which can affect overall project performance in the construction industry.

The practical application of the findings is relevant to construction professionals, construction stakeholders, and government agencies on decision-making and strategies to improve construction projects through the early identification of pertinent construction constraints that might affect construction project performance. It will also expand stakeholders' knowledge and understanding of the impact of constraints' determinants such as cost, time, risk, scope, quality, and resources on projects' success. It will assist stakeholders in identifying and overcoming construction project execution constraints. This study is limited by the inability of the authors to explore more case studies involving a large sample.

The study also recommends that the six components of construction constraints identified in this study should guide professionals in the construction industry in effectively improving the performance of construction projects. However, due to time constraints, the study was limited to construction professionals within Free-State province, South Africa, showing that the findings cannot be generalised for the South African construction industry. Nonetheless, it is important to note that construction professionals in Free-State Province used for this study account for the substantial professional activities in the South African construction industry. Equally, future studies can be conducted to test the six components identified in this study as they affect the traditional concept of triple constraints (time, cost, and quality) in construction project performance. This can be done by incorporating all built environment professionals with first-hand experience in construction project planning, execution, and management.

**Author Contributions:** Conceptualisation, B.F.O. and T.G.M.; methodology, T.G.M., B.F.O. and C.O.A.; resources, T.G.M. and B.F.O.; writing—original draft preparation, T.G.M. and B.F.O.; writing—review and editing, B.F.O., C.O.A. and B.O.A.; visualisation, T.G.M., B.F.O. and B.O.A.; supervision, B.F.O., C.O.A. and B.O.A.; project administration, B.F.O., C.O.A. and B.O.A. All authors have read and agreed to the published version of the manuscript.

**Funding:** This research received no external funding.

**Institutional Review Board Statement:** Not applicable.

**Informed Consent Statement:** Not applicable.

**Data Availability Statement:** Not applicable.

**Acknowledgments:** We acknowledge the efforts of the peer reviewers in helping to improve the quality of the article with their constructive comments.

**Conflicts of Interest:** The authors declare no conflict of interest.

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
