# Peer review of "Assessing Construction Constraint Factors on Project Performance in the Construction Industry"

_buildings, doi:10.3390/buildings12081183_

Round 1

Reviewer 1 Report

I have read the manuscript and have the following observations:

there have been several studies that look on the project contraints e.g.:

Deep, Shumank, Shankar Banerjee, Saurav Dixit, and Nikolai Ivanovich Vatin. "Critical Factors Influencing the Performance of Highway Projects: An Empirical Evaluation." Buildings 12, no. 6 (2022): 849.

can the author comment on how their work adds the message sent out by these?

Furthermore, i would like to recommend that the authors should include more statistical details to re-write their introduction. 

Authors should discuss how factor analysis is helpful in achieving their aims

Lastly i want the authors to re-write the discussion and  conclusion section to improve it readability

Author Response

REVIEWER 1

Comments and Suggestions for Authors

I have read the manuscript and have the following observations:

there have been several studies that look on the project contraints e.g.:

Deep, Shumank, Shankar Banerjee, Saurav Dixit, and Nikolai Ivanovich Vatin. "Critical Factors Influencing the Performance of Highway Projects: An Empirical Evaluation." Buildings 12, no. 6 (2022): 849.

can the author comment on how their work adds the message sent out by these?

Response

Has requested this has been included in the introduction of this paper.

Furthermore, i would like to recommend that the authors should include more statistical details to re-write their introduction. 

Response

This has been addressed in the introduction part of the manuscript

Authors should discuss how factor analysis is helpful in achieving their aims

Response

The benefit of using EFA in achieving the study's aims has been included in the introduction part of the manuscript.

Lastly i want the authors to re-write the discussion and conclusion section to improve it readability.

Response

The discussion and conclusion part has been re-written to improve their readability

Reviewer 2 Report

·       Need proofread- minor spelling mistakes and wrong citation style found in the text.

·       Referring to the methodology:

-how do authors identify the population of this study, which stated as 27, 000.

- Justification on the selection of province, Free State.

·       Respondents’ professional qualification - should be Figure 3.

·       Figure 2: Respondents’ professional qualification – typo on the project management.

Author Response

REVIEWER 2

Comments and Suggestions for Authors

  • Need proofread- minor spelling mistakes and wrong citation style found in the text.

Response

This has been corrected throughout the manuscript

  • Referring to the methodology:

-how do authors identify the population of this study, which stated as 27, 000.

- Justification on the selection of province, Free State.

 Response

This has been addressed in the methodology section of the manuscript

Respondents’ professional qualification - should be Figure 3.

Response

This has been corrected

 Figure 2: Respondents’ professional qualification – typo on the project management.

Response

This has been corrected

Reviewer 3 Report

I think the term constraint should be more clearly described, and I wonder if that is the right word to use in the title or in this research.

In the introduction, research on factors causing delay and cost overruns among others are used to describe factors causing constraints. Is factors causing constraints right word? More place should be used discussing the purpose of the research and possible research questions in the introduction. 

The literature review is unprecise when it comes to the use of the word constraints. Again, different literature from factors causing delay is used to study the factors causing constraints. The words should be defined and used in more clearly way. It is not clearly described how Table 1 is developed. 

Methodology should be better described. In the abstract it is mentioned that this is a case study, but a case study is not mentioned in the methodology section. A discussion of the choice of methodology is missing. It is not clearly described how the survey or the questionnaire was performed. How was the questionnaire set up, what were the questions? 

The result section presents the results in an ok way, but the quantitative results should be followed by an explanation. I think some of the tables or figures are not needed. More explanation is needed. Table 4 is not well organized and unclear. The results are gathered in 6 main components, but it is unclear how and why this is done. 

The conclusions should be more clearly described. A clear research objective would help to be more specific on the conclusion. What is the main contribution to research and to practice? 

Author Response

REVIEWER 3

Comments and Suggestions for Authors

I think the term constraint should be more clearly described, and I wonder if that is the right word to use in the title or in this research.

 Response

The term constraint has been more clearly described in the manuscript's introduction.

In the introduction, research on factors causing delay and cost overruns among others are used to describe factors causing constraints. Is factors causing constraints right word? More place should be used discussing the purpose of the research and possible research questions in the introduction. 

Response

This has been addressed in the introduction aspect of the manuscript

The literature review is unprecise when it comes to the use of the word constraints. Again, different literature from factors causing delay is used to study the factors causing constraints. The words should be defined and used in more clearly way. It is not clearly described how Table 1 is developed. 

Response

This has been addressed in the literature review section of the manuscript

Methodology should be better described. In the abstract it is mentioned that this is a case study, but a case study is not mentioned in the methodology section. A discussion of the choice of methodology is missing. It is not clearly described how the survey or the questionnaire was performed. How was the questionnaire set up, what were the questions? 

Response

This has been addressed in the methodology section of the manuscript.

The result section presents the results in an ok way, but the quantitative results should be followed by an explanation. I think some of the tables or figures are not needed. More explanation is needed. Table 4 is not well organized and unclear. The results are gathered in 6 main components, but it is unclear how and why this is done. 

Response

Table 4 has been corrected as requested. Also, the reason why the result is gathered in 6 main components is explained under the pattern matrix discussion.

The conclusions should be more clearly described. A clear research objective would help to be more specific on the conclusion. What is the main contribution to research and to practice?

Response

This has been included in the conclusion section

Reviewer 4 Report

See attached file for comments

Author Response

Reviewer 4

See attached file for comments

Response

Response: All the comments highlighted in the attached document have been addressed.

Round 2

Reviewer 3 Report

My comments from the first review round are still valid. I don't think that the authors have fixed my previous concerns. Some comments in the attached document support this. The word constrains is still not well explained and still mixed with delays etc... 

I did not comment on the last part of the paper as long as the first part of the paper is not fixed and is not well describing the background of the research and the state of the art. 

Author Response

Reviewer's 3

Ref 22 doesn't menion constraints at all. This reference chek how characteristics of project based firms and how these characteristics influence how they manage innovations. How is this relevant in this setting?

Response

The statement has been rephased in the text, and appropriate citation has been used.

causing delay! This article doesn't mention constraints at all... It states that delay in construciton projects lead to disputes and litigations. How is this relevant for your research?

Response

The statement has been rephased in the text, and appropriate citation has been used

I can not find this in Ref 24. What does it mean that constrains could involve all the parties in a contract?

Response

The statement has been rephased in the text, and appropriate citation has been used and listed

how does it include time and cost overrun?

Response

The statement has been corrected

Ref 28 doesn't mention constrints at all.

Response

The statement has been rephrased in the text, and the citation has been corrected and listed.

I still don't think that the introduction build up to the demand of a research study focusing on identifying factors causing constraints. Some of the sources that are used as an argument for this don't mention conatraints at all, as I have already mentioned in the above comments.

Response

Corrections on all sources pointed out have been carried out

ciriticality of factors that invluence the performance of projects, is that what you are aiming for? That is something else than factors causing constraints, isn't it?

 Response

The first reviewer has asked us in his/her comment that we should compare our study with Ref [30 ] which is now Ref [29]

  It is not well explained, so the identification of a research gap is weak described in the introduction.

Response

All concerns areas leading to the gap in the introduction part have been addressed as suggested.

which studies?

Response

This has been addressed, and appropriate reference has been provided and listed

belong to the methodology section. How have you shown that his method is not previously used?

Response

The statement has been rephrased

So the constrains factors are time, scope, qualiyt, cost, resources, risks. You are looking for factors causing conatraints, meaning factors causing cost, time, scope, .... How is this linked together? I don't understand your meaning of factors causing constraints......

 Response

The statement has been rephrased.

 causing project delay, not constraints...

As previously commented, the source doesn't mention constraints...

Response

Both the text and citation have been corrected

is it in the ref used delay/constraints or only delay?

Response

Both the text in the manuscript and the reference have been corrected

causing delay, not causing constraints...

Response

The text has been rephrased  

As mentionaed in the last review, Ref 52 doesn't mention these five categories at all... Where are these five categories from

Response

The reference has been corrected in line with the statement and also replaced in the reference list

sure?

Response

Acknowledged

why? And how?

Response

The statement has been recast to suit the methodology flow of the manuscript.

Reviewer 4 Report

The corrections in the text make it much easier to understand your contribution to the field. The attached file contains a few last modifications for the English review.

Author Response

Reviewer 4

Response: All the comments highlighted in the attached document have been addressed

Round 3

Reviewer 3 Report

I have started the third round of review of this paper. My comments in the first two rounds show clearly that I don’t agree in the concept of the paper and the use of the word constraints. The authors have fixed some of the wrong use of previous literature, but have not done something with their concept. Meaning that they still are searching for factors causing constraints, and give a list of factors, based on the literature review. But in the literature, this is not explained as factors causing constraints, but in most cases, factors causing delay, sometimes factors causing conflicts, etc... I think I will not be able to agree in the authors concept, and they will not be able to adapt their paper to my comments. I see that the other reviewers have not raised the same concerns with the paper as I have, so maybe it is just maybe I that cannot follow their logic in the article. In my opinion, when going quickly through the new version, they have done minor changes, while what I was asking for was major changes. But if I should comment once more, I would give some of the same overall comments all over again. So I think either you should listen to the three other reviewers and accept the article with maybe minor revisions. Or according to my comments (being a critical reviewer) I would say that the authors are not able to do the changes I ask for and then I would say that reject would be a possible outcome. This is a hard decision of course, but the authors have had a chance in two rounds now to see my main points, but they have not taken it into consideration.